# Relationship between Executive Functions, Social Cognition, and Attachment State of Mind in Adolescence: An Explorative Study

**DOI:** 10.3390/ijerph20042836

**Published:** 2023-02-06

**Authors:** Riccardo Williams, Silvia Andreassi, Marta Moselli, Fiorella Fantini, Annalisa Tanzilli, Vittorio Lingiardi, Fiorenzo Laghi

**Affiliations:** 1Department of Dynamic and Clinical Psychology, and Health Studies, Faculty of Medicine and Psychology, Sapienza—University of Rome, Via degli Apuli 1, 00185 Rome, Italy; 2Department of Developmental and Social Psychology, Faculty of Medicine and Psychology, Sapienza—University of Rome, Via dei Marsi 78, 00185 Rome, Italy

**Keywords:** attachment, executive functions, social cognition, adolescence

## Abstract

Background: The passage from pre-adolescence to adolescence is presented as a turning point for the achievement of those abilities in social understanding as they commonly appear in adulthood. Developmental perspectives point to the possible role of neuro-cognitive maturation and social experiences to facilitate this growth. This paper has the goal to goalsto propose a valid and reliable measure of the new quantitative and qualitative advancements in social understanding occurring in the adolescent passage; relying on this, the research has two main objectives (a) to establish the associations between the advancements in social understanding and the executive functions held responsible for the neuro-cognitive rearrangements of adolescence; (c) to evidence the significant associations between attachment models and the development of social understanding in this phase of life. Methods: 100 subjects (50 boys and 50 girls, aged 11–15) were administered with AICA, SCORS, CNT, Stroop Color-Word Test, and WISC-III. Results: Advancements in the complexity of self–other representations and mentalization of interpersonal exchanges significantly occurs in the passage from pre-adolescence to adolescence and seem to be promoted by increased performances in executive control and cognitive shifting. Dismissing state of mind with respect to attachment is associated with lower development of social understanding in adolescence. The neurocognitive reorganization that underlies the passage from pre-adolescence to adolescence seems to provide the scaffolding for more sophisticated interpretations of the social world. Past and current affective experience can boost or hinder the full deployment of such human maturational potential. Given the importance of social cognition for adjustment and psychopathology, clinical intervention should target the amelioration of individual and family abilities in social reasoning and mentalization.

## 1. Introduction

In the developmental perspective, many aspects of social interactions are thought to be oriented by a series of underlying competences considered in a specific domain of functioning referred to as social cognition [1,2]. Empirical research has evidenced the importance of attachment security for the development of healthy adjustment, with specific reference to the quality of social relationships and social understanding in the life span [3,4,5] and in particular in adolescence [6,7]. The neuroscientific perspective applied to this filed of developmental theories have also evidenced that social competences evolve with a specific sequence and are possibly influenced by maturational factors [8]. In this paper, we shall analyze the relative impact of attachment experiences (models) and maturation of executive functions on the growth of those aspects of social understanding expected to evolve in the critical passage from pre-adolescence to adolescence.

### 1.1. The Development of Social Cognition

Social cognition is broadly defined as an area of cognition in which people perceive, think about, interpret, categorize, and judge their own and others’ social behaviors [9]. It is considered to ensue from a set of specific abilities that allow to understand the mental states or psychological dispositions supporting interactions and individual behaviors and has been employed as a useful construct for the study of social functioning and other relevant individual differences in overall adjustment [1].

Social cognition is commonly held to derive from some innate human skills to interpret the social world [10]. However, more mature forms of social understanding are otherwise shown to be the developmental outcome of some important phases of psychological reorganization, among which the passage from pre-adolescence to adolescence is considered a critical period [2,11,12]. Developmental theorists have pointed out that as children mature, they progressively reach an increased capacity to distinguish the points of view of self and others in a more clearly and articulated way [13]. As far as adolescence is concerned, some specific skills are reported to significantly increase in the passage from the previous periods of development [14,15,16]. For instance, the skill for coding complex emotional expressions undergoes a final refinement in the first years of adolescence, while the capacity to differentiate perspective taking continues to develop through late adolescence into adulthood [17,18]. 

Many authors consider the impact that the progressive quantitative changes occurring in late childhood and adolescence in these single skills underpin an overall qualitative change in the area of social understanding [19]. Developmental researchers have had difficulties in operationalizing these aspects of social understanding in adolescence into specific quantitative variables. However, observational studies have highlighted that the supposed general advancements in social reasoning result in a new and more sophisticated form that now more closely resembles the features of the adult functioning in this area. 

In the first place, any interpretation of human behaviors relying on the category of psychological causality seems to be finally established in adolescence [20,21]. Adolescents have increasingly sophisticated abilities that generate a recursive social reasoning into which simple as well as complex states of mind and personal dispositions are regarded as the causes that stay behind individual initiatives and any aspect of human interactions [21,22,23,24]. In this regard it has been noted that adolescents seem to develop a stable awareness of how each social agent’s internal state and consequent behavior is influenced by the understanding of the other’s internal state and psychological behavior, which is, psychological reciprocity or explicit intersubjectivity [25]. The new and more advanced aspects of adolescent social reasoning are nurtured by more complex and articulated self–other representations. These representations are now less linked to behavioral outcomes and contingent consequences of interactions, recognize and differentiate more stable personality dispositions from more transient attitudes toward the external reality, integrate conflicting and ambivalent tendencies into a unique personal identity, and possibly include individual unconscious dispositions [13,26].

All developmental approaches presuppose that the critical reorganization in social reasoning occurring in the course of development, including adolescents, is influenced by past and current factors, focusing on both interactive experiences and neuro-cognitive maturation [27]. Some relevant hypotheses have been proposed in the respective fields of study, but little research has addressed the relative weight of specific environmental and maturational variables on measurable increases in social reasoning in this sensible period of development [28,29,30,31].

### 1.2. Attachment and Social Interactions

Attachment theory has proposed that Internal Working Models (IWMs)of attachment derive from the organization of early affective experiences into schemes of expectations concerning the caregivers’ emotional availability and responsiveness as well as the sense of the self as being worth of care and love [32]. IWMs are also thought to give shape to orientate social interactions with meaningful others in the future [33,34].

Longitudinal studies in this field evidenced that the various IWMs (early relational representations and expectations of early relational experiences) evolve in a lawful way and organize the future interactions in the various social contexts according to what Bowlby posited. Empirical research showed that attachment models as evaluated through the Strange Situation [33] allow the prediction of the individuals’ quality and types of social behaviors in different developmental context, such as playful interactions in pre-school years [4], relationships with teachers in school years [35], group and family transactions in adolescence [7,25,36,37], and romantic relationships and couple dyadic interactions in adulthood [35]. Attachment theory also contemplates the potential developmental plasticity of IWMs so that meaningful changes in social interactions or life events can contribute to modify early representational models of affective interactions [38,39]. On the other hand, when no such significant changes in the interactive experiences occur during development, the IWMs tend to stay stable through the lifespan. In a more recent perspective, attachment system has been conceived of as the ideal environment of evolution of the human capacity to attribute mental states and interpret behaviors in a mentalistic perspective [40,41]. Individual differences in the attachment models have then been reinterpreted to consist of diverse modes representing intentional dispositions underlying self–other interactions [42].

Recent theoretical interpretations and empirical evidence have indeed highlighted how the specific associations between attachment IWMs and individual differences in social interactions are mediated by differences in social understanding [37,40,43,44,45]. However, the accruing empirical evidence confirming the link between attachment models and social cognition is mainly limited to clinical samples [2,3,4,46]. A possible influence of attachment models and individual differences in social cognition have also been proposed with reference to the adolescent period [7,25,47]. To the best of our knowledge, no empirical investigation has been carried out as to the influence of attachment models on the reorganization of social cognition occurring in the passage from pre-adolescence to adolescence in normative samples.

### 1.3. Executive Functions and Social Cognition in Late Childhood and Adolescence

Neuroscientific research has highlighted the essential aspects of the brain maturation possibly underlying the cognitive and socio-cognitive development of adolescence [5,6,7,8,9,48,49]. The important growth of social cognition in adolescence has been linked to the profound neuro-cognitive reorganization occurring in this phase [50,51]. 

Even if relatively little is known about the development of executive functions during the school-age years and around the transition to adolescence [52], there is a growing evidence that some executive functions, above all the ability to inhibit the first response and to produce an alternative response and cognitive shifting, may account for some important psychological changes occurring from childhood during adolescence [53,54].

With regards to inhibition functions, Garon, Smith, and Bryson [55] distinguished simple from complex response inhibition tasks based on whether working memory is also needed. Simple response inhibition, such as go-no-go-like tasks, requires a minimal amount of working memory and shows its rudiments during infancy. Complex response inhibition, such as Stroop-like tests, requires substantial working memory, because the child has to inhibit one response and produce an alternative one and continues to develop throughout adolescence, lagging behind the development of other cognitive skills [56,57,58]. The literature suggests that the developmental trajectory of cognitive processing needed for the Stroop task is characterized by the increasing ability to recruit additional frontal neural resources. This is supposed to explain why adolescent development of executive functions has a significant impact on general cognitive reorganization of the phase [59]. 

Cognitive shifting involves the ability to perform a new operation in the face of a proactive interference or negative priming [60]; the ability to inhibit previously activated mental sets and the maintenance and updating of that mental set based on feedback (working memory) seem to be prerequisite processes for successful performances at shifting tasks [61]. This capacity is thought to be one of the last executive functions to fully mature [62], and complex tasks show further development in older children and adolescents [57,63].

Davidson and colleagues [64] found improvement from infancy through adolescence; in particular, Manly and colleagues [65] and Taylor and colleagues [66] found a significant enhancement in cognitive shifting between 11–13 and 14–15 years of age [65].

The neuro-scientific literature indicates that the complexity of social cognitive competences (e.g., attribution of intentions and other mental states) reached in adolescence hinges on the activation of the same areas recruited for executive control [67]. In this sense, it has been hypothesized that the growth of executive control promoted by the maturation of pre-frontal cortex can at least partially account for the parallel development of social cognition [68,69].

### 1.4. Aims

In keeping with the literature concerning the development of social cognition, executive functions, and attachment in adolescence this study aims at answering at two questions.

(a)Is the increase in the measure of the complexity of social reasoning in adolescence significantly associated with the increase in the performances at executive tasks proved to mature in adolescence, namely, inhibitory control and cognitive shifting?(b)Are the models of attachment, as assessed through the evaluation of the States of Mind with Respect to Attachment, significantly predictive of the measure of complexity of social reasoning in adolescence?

In order to answer these two research questions, the construction of a reliable and valid measure of social understanding in the window of development was also carried out.

## 2. Materials and Methods

### 2.1. Participants and Procedure

The sample consisted of 100 adolescents (50 boys and 50 girls). The average age of the students was 13.22 (SD = 1.57; range 11–15). For comparative purposes the sample was divided into two age groups: 11–12 years (N = 43; M = 21; F = 22) and 13–15 years (N = 57; M = 29; F = 28). Before the data collection, letters describing the study were sent to school principals and parents, and the participation only took place after signed consents from parents were returned. Participation rates ranged from 95 to 100 percent in classrooms. Each adolescent was tested individually in a quiet school area by a female, trained research assistant for a total of approximately 180 min over three occasions in a quiet school area provided by the principals. On the first occasion, students were tested with a measure of verbal intelligence and executive function tasks, whilst on the second occasion, the Attachment Interview for Childhood and Adolescence (AICA) was administered, and on the third occasion the Thematic Apperception Test was administered. This survey was reviewed and approved by the Ethics Commission of the Department of Dynamic, Clinical Psychology and Health Studies of “Sapienza”, University of Rome.

### 2.2. Measures

*State of Mid with Respect to Attachment* The Attachment Interview for Childhood and Adolescence (AICA; Ref [70] is a revised version of AAI (Adult Attachment Interview) for participants in early adolescence and adolescence. The contents are similar to those of the AAI, but some questions have been linguistically simplified and others added to capture specific characteristic of attachment in adolescence. The AICA is an hour-long, semi-structured interview and is focused on the interviewee’s relationships with attachment figures during childhood and early attachment experiences. According to Ammaniti et al. [70], the AAI coding system [71] appears to be reliably adaptable to the AICA material, and a high degree of intercoder agreement can be reached with well-trained AAI coders, as confirmed in other studies [72,73]. The coding system developed by Main and Goldwyn (1998) and used in an Italian context by Ammaniti et al. (1990) was used to classify adolescents into one of four categories for overall state of mind with respect to attachment: (a) secure, freely valuing of attachment; (b) dismissing of attachment relationships; (c) preoccupied with attachment relationships; and (d) unresolved with respect to past loss or trauma. In addition to these classifications, the interview transcripts were scored on seven 9-point scales that assessed the present mental representation of attachment (idealization, anger, derogation, passivity, coherence of transcript, lack of memory, and coherence of mind). According to Main and Goldwyn [71] and Ammaniti and colleagues [70], the scales that were scored separately for mothers and fathers (idealization, anger, and derogation) were combined, and the overall score on these representational scales were used in the analyses. Interviews were audiotaped and later transcribed verbatim, retaining all stuttering, grammatical errors, and mispronunciations and marking any interruptions and pauses. Coding of the AICAs in this sample was conducted by coders trained for reliability in the use of the AICA classification system [71]. One half of the transcripts were double-coded. Disagreements between two coders on transcripts in the present study were resolved by conferencing with a third coder. Coders were unaware of all hypotheses and had no knowledge of participants responses on the other instruments. Exact agreement between two raters on the overall three-way attachment classification was 86% (k = 0.73).

*Social Cognition*. The Social Cognition and Object Relations Scale (SCORS; 77) was used to evaluate participants’ quality of object relations from TAT responses. The SCORS consists of 4 variables that are scored on a five-point anchored scale in which lower scores (e.g., 1 or 2) indicate more pathological responses, and higher scores (e.g., 4 or 5) indicate healthy responses. More thorough descriptions of the five SCORS variables and global rating method are provided in the manuals developed by Pace and colleagues [74] and by Abbate and Massaro [75] for Italian adaption. The two pertinent scales for evaluating social cognition were Complexity of Representation of the people (COM), which evaluates how well the participant is able to see internal states in the self and other when reporting narratives as well as the ability to integrate both positive and negative aspects of self and others, and Understanding of Social Causality (SC), which assesses the extent to which the participant understands human behavior. For each subscale, a score ranging from zero to five is applied to the answers provided by each subject to the TAT tables. The SCORS variables were dimensionally scored based on relational episodes and self-statements verbally expressed directly to the female trained during the course of the evaluation. One half of the transcripts were double-coded. Disagreements between two coders on transcripts in the present study were resolved by conferencing with a third coder. Coders were unaware of all hypotheses and had no knowledge of participants responses on the other instruments. Reliability for two coders was calculated by correlating their scores on COM and SC scores for each interview. Range reliability was 0.86–0.88 and was calculated using the Spearman Brown correction formula. Disagreements were resolved by conferencing. The correlation between COM and SC scores was high (r = 0.79, *p* < 0.001). We combined the scores by averaging them, yielding a global measure of social cognition. Higher scores indicated higher social cognition.

*Cognitive Shifting.* We used the Contingency Naming Test (CNT) [66] to assess cognitive flexibility under moderate working memory demands. It includes four subtests, each one increasing in difficulty level. The participant is presented with an A4 card on which are printed three rows of shapes of different colors (pink, blue, or yellow). Within each outside shape, a second inside shape is drawn (shapes include circles, squares, or triangles). Above some of the stimuli, a reverse arrow is drawn. The CNT requires naming the color or shape of a series of stimuli according to different rules. Subtests 1 and 2 ensure the participant can name the colors and shapes, and they act as baseline measures of selective attention and processing speed and. These tasks serve as a warm-up to the switching task introduced by Level B rules. Level B rules present working memory and inhibitory control demands because they require switching between naming the stimuli by color or by shape, depending on one attribute or two attributes. For each CNT task, subjects saw one row of 9 stimuli on an 8” x 11” sheet of white paper, which they verbally labeled during an un-timed warm-up trial. Once it was established that the rule had been learned, subjects were presented with three rows of 9 stimuli on an 8” x 11” sheet of white paper. Subjects named the stimuli as quickly as possible while the experimenter recorded reaction time (RT) using a handheld stopwatch. As reported by Halberda, Mazzocco, and Feigenson [76], the variable of interest is performance efficiency, measured by assessing the speed–accuracy trade-off via the following formula: Efficiency = [(1/RT)/√(errors + 1) ] × 100. Higher scores indicated higher cognitive flexibility.

*Inhibitory control*. We also administered the Stroop Color–Word Test [77] that includes different tasks, consisting of 30 words in 3 consecutive lines: (1) naming words that are printed in black ink on a card (W); (2) naming the color words that are printed in color ink (C); (3) naming the color words that are written in a different color (CW). The time taken to complete parts of the test was measured as reaction time. Stroop interference, measured by the difference in latency between the different trials, is typically taken to reflect inhibitory ability, according to the following formula: CW − [(C+W)/2]. Higher scores indicated lower inhibitory control.

*Verbal ability*. We used the vocabulary subtest from the Italian version of WISC III [78] to assess word knowledge, language development, and long-term memory. This task measures lexical knowledge and, more specifically, the ability to retrieve a word’s meaning and to provide an accurate definition. Participants were asked to orally define 30 stimulus words presented by the examiner; the words increased in difficulty and abstraction as the test progressed. Each child received zero, one, or two points for each item based on the quality of the definition given. The score can range from 0 to 60.

### 2.3. Data Analysis

All data analyses were conducted using SPSS Statistics Version 22. Preliminarily, we tested sex, age, and verbal differences, because for these variables there were some potential covariates identified for the analyses involving the executive functioning and social cognition scores. To investigate sex and age differences, we conducted a factorial ANOVA on executive function dimensions (CNT and the Stroop Color–Word Test) and factorial ANCOVA for social cognition scores, using verbal ability as covariate. Chi-square analyses were used to verify association between sex, age, and AAI classification. An analysis of covariance with AAI classification as a between-subjects factor was conducted for executive function dimensions (CNT and the Stroop Color–Word Test), using age as covariate, and for social cognition score, in which age and verbal ability were used as covariates. Partial eta-squared values were calculated as a measure of effect size, and results were interpreted using Cohen’s (1988) guidelines for determining small (0.01), medium (0.06), and large (0.14) effects.

Pearson correlations were performed to examine the relationship among the key variables: attachment state of mind dimensions, executive function dimensions, and social cognition. 

## 3. Results

### 3.1. Preliminary Analyses: Sex and Age Differences on Executive Functions and Social Cognition Dimensions

The univariate distributions of the study variables were examined for normality (i.e., via skewness and kurtosis values and the Kolmogorov–Smirnov statistics). These variables were normally distributed. 

Descriptive statistics for the key variables used in the present study are presented in Table 1. Factorial ANOVA on CNT scores revealed significant main effects of age, F (1.98) = 30.43, *p* < 0.001, ηp^2^ = 0.28, but not for sex, F (1.98) = 0.67, *p* = 0.41. There was no significant effect of the interaction between sex and age, F (1.98) = 0.31, *p* = 0.58. These results evidenced that 13–15 years-old adolescents obtained higher scores than 11–12 years-old adolescents. Similarly, factorial ANOVA on Stroop Test scores revealed significant main effects of age, F (1.98) = 16.35, *p* < 0.001, ηp^2^ = 0.15, but not for sex, F (1.98) = 0.08, *p* = 0.78. There was no significant effect of the interaction between sex and age, F (1.98) = 0.06, *p* = 0.81, confirming the developmental trend evidenced with cognitive flexibility scores.

The factorial ANCOVA, using verbal ability as a covariate on social cognition (CR and SC combined scores), revealed significant main effects of age, F (1.98) = 23.62, *p* < 0.001, ηp^2^ = 0.20, and for sex, F (1.98) = 5.52, *p* < 0.01, ηp^2^ = 0.05. There was no significant effect of the interaction between sex and age, F (1.98) = 1.73, *p* = 0.19. Verbal ability effect was not significant, F (1.98) = 0.11, *p* = 0.74. These results demonstrate that girls obtained higher scores than males and that 13–15 years old adolescents obtained higher scores than 11–12 years old adolescents, as reported in Table 1.

### 3.2. Attachment Classification: Sex and Age Differences

The distribution of the three attachment classification groups is presented in Table 2. The 100 AICAs were classified as follows: 67 (67%) secure, 23 (23%) preoccupied, and 10 (10%) dismissing. No subjects in the study were classified as unresolved with respect to attachment. Males and females were equally distributed among attachment classification groups, (2) = 0.58; *p* = 0.75. Chi-square analysis revealed age differences among the attachment groups, (2) = 9.65; *p* < 0.001. As displayed in Table 2, 13–15 years old were more likely to be classified as secure (78.9% of 13–15 years old male vs. 51.2% of males; R = 1.1), and 11–12 years old adolescents were more likely to be classified as dismissing (37.2 % of 11–12 years old vs. 12.3% of 13–15 years old; R = 1.9).

### 3.3. Attachment Classification, Executive Function Dimensions, and Social Cognition

The ANCOVA (see Table 3), using age as covariate, on CNT scores did not reveal significant main effect of classification attachment group, F (2.97) = 0.11, *p* = 0.89. The age, as a covariate, had a significant effect, F (1.98) = 21.80, *p* < 0.001, ηp^2^ = 0.18. For the Stroop test, ANCOVA revealed significant differences for attachment classification, F (2.97) = 3.78, *p* < 0.01, ηp^2^ = 0.07. The age, as a covariate, had a significant effect, F (1.98) = 12.38, *p* < 0.001, ηp^2^ = 0.11. Post hoc comparisons (Tukey Test, *p* < 0.05), after adjusting for age, revealed that the secure and preoccupied groups, that did not differ, obtained lower scores (high inhibitory control) than the dismissing group. 

The ANCOVA revealed significant differences on social cognition scores for attachment groups, F (2.97) = 7.63, *p* < 0.001, ηp^2^ = 0.14. Verbal ability had not a significant effect, F (2.97) = 1.67, *p* = 0.20. Post hoc comparisons (Tukey Test, *p* < 0.05) revealed that secure and preoccupied group, which did not differ, obtained higher scores than the dismissing group.

### 3.4. Relationship between Attachment State of Mind Dimensions, Executive Function Dimensions, and Social Cognition

Pearson correlations were performed to examine the relationship among the key variables considered in the present study: attachment state of mind dimensions, executive function dimensions, and social cognition.

Social cognition was significantly correlated with executive functions (CNT and Stroop Test scores) and with the scales describing the state of mind with respect to attachment assessed through the AICA. In particular, the scales (see Table 4) associated with the dismissing state of mind with respect to attachment show a significant negative correlation with social cognition: lack of memory r = −0.43, *p* < 0.01 and idealization r = −0.46, *p* < 0.01; the scales associated with the preoccupied state of mind passivity are significantly positively correlated with social cognition, r = 0.25, *p* < 0.05; the scales of coherence of transcript and coherence of mind, associated with the free-autonomous state of mind, are significantly positively correlated with social cognition: r = 0.28, *p* < 0.01 and r = 0.29, *p* < 0.01, respectively.

## 4. Discussion

In this research, we aimed at describing the qualitative and quantitative growth of social understanding in the passage from pre-adolescence to adolescence and to analyze the relative contributions of neuropsychological and attachment experiences to the advancements in social cognition in this developmental phase.

As a first step of the research, the construction of a reliable and valid measures systematically assessing social understanding in adolescence was carried out. As indicated in the introduction, the combination of quantitative and qualitative measures of social understanding employed in this cross-sectional study encompasses two aspects of social understanding considered by both developmental and clinical observations [19,78]. The employment of such measure seems to allow to systematically capture the nature of the qualitative changes occurring in social reasoning in adolescence. Indeed, the qualitative and quantitative advancements concerning both the complexity and articulation of personal representations and mentalistic understanding seem to proceed with age, regardless of gender differences as described at an anecdotal level in the literature [79]. At the same time, it has to be noted that in our sample the combined measures of social understanding evidenced a significantly higher level of functioning in girls regardless of age. Our results are in line with previous evidence about gender differences in social cognition skills, since both in previous stages of development as well as in adulthood females fare better off than males in tasks of social cognition [80,81].

The second objective of this study was to verify whether the significant increase in the measure of the complexity of social reasoning was significantly associated with the increase in the performances at those executive tasks that mature in adolescence, namely inhibitory control and cognitive shifting. Assuredly, our results evidenced that the positive correlation between age and the core for complexity of social reasoning was accompanied by a concurrent significant association between age and the two inhibitory controls and cognitive shifting. Even more clearly, the ANOVA models indicate that better performances at inhibitory control and contingent shifting tasks are associated with social understanding. Furthermore, performances at STROOP and CNT tests are not significantly influenced by sex differences.

Overall, this set of results highlights how at least part of the observed increase in the complexity of social understanding determined by age is accounted for by the parallel neuropsychological growth in executive functions. It is noteworthy that the improvement of executive control is regarded by the current neurocognitive literature as the neurobiological underpinning of the more general cognitive rearrangement through adolescence [19,82,83,84] to adulthood [85]. Clear theoretical links can also be traced between the single neurocognitive abilities tested in this paper and the specific areas of social understanding that showed an improvement in our sample. In particular, it is believed that the increase in self-reflection is warranted by the maturation of cognitive flexibility and abstract thinking measured through the CNT tasks. The increased cognitive flexibility registered in the older boys and girls would nurture the typical recursive quality of thinking that characterizes adolescence [78]. Evidently, the increased capacity to monitor one’s own behavior and thinking also leads to a more flexible and articulated model of the experience of relationships. This enhanced monitoring allows boys and girls alike to encompass ambivalent aspects of relations into a dynamically fluent image of themselves and others, which is the complexity of representation [13]. The capacity to flexibly consider different mental causes of behaviors at the same time allows them to differentiate more stable individual personality traits from dispositional traits.

As far as inhibitory control is concerned, several studies have shown that this executive function is required for perspective taking [86]. The literature on the theory of the mind showed that a capacity for self–other differentiation and intersubjective connection can be established only if the inhibition of the default mode of self-centered interpretation of interactions is achieved [87,88]. This study seems indeed to point to the relevance of the consolidation of inhibitory control occurring in the course of adolescence for the possibility to further develop an intersubjective exchange from this phase of development on.

As far as the third objective of the study is concerned, the results outline an interesting relationship between attachment models and the development of social understanding. The dismissing state of mind with respect to attachment and the relative sub-scales of the state of the mind of lack of recall and idealization show a negative impact on the evolution of articulated and complex interpretation of the interpersonal world. Remarkably, in this sample, dismissing attachment is significantly associated with lower scores in the social cognition measure independent of the maturational factor represented by age. It could be suggested that the affective core of attachment models is able to exert a negative influence on social cognition owing to their relatively stable affective predictions that shape the move to the level of representation [33,89,90]. In truth, the subscales of lack of recall and idealization as well as the overall representational strategy characterizing dismissing individuals is thought to prevent the exploration and articulation of thoughts and discourse concerning interactive experiences. Dismissing reports of affective experiences tend to be poor in quality, superficial, overly concrete, bereft of psychological causal interpretations, and lacking any reference to negative aspects of interaction [90,91]. We contend that this representational and conversational strategy may have a significant impact on the development of complex and articulated social understanding as well as on the mentalistic comprehension of agency and interactions relevant for the development of social understanding in adolescence. Dismissing attachment has been demonstrated to correspond to specific conversational strategies adopted by the adolescents and their parents in tasks entailing the discussion of interpersonal conflicts and points of agreement [92]. The conversational strategies corresponding to dismissing AAI classifications evidence a very limited capacity for perspective-taking and explicit intersubjective sharing [93].

Another interesting aspect to discuss concerns the lack of a significant link between the preoccupied model and impaired social understanding. In this study, individuals classified as preoccupied seem to better free-autonomous individuals at the social cognition scores. This result seems to be tally with other studies indicating that preoccupied states of mind with respect to attachment may entail a continuous attempt at controlling others’ mental states [94,95,96,97,98]. Our results may therefore be considered as a result of the preoccupied individual to hyper-mentalization, resulting in excessively articulated representations and psychological interpretations of the social world.

We are aware that the cross-sectional design as well as the size of the sample of this study lend only limited support to the incremental value of executive function for the development in adolescence. It should anyway be noted that the measures of inhibitory control and cognitive shifting chosen included in this study were chosen because of the steady progress they proved to show in the age span considered for our sample. 

## 5. Conclusions

This study highlights the importance of the neuro-cognitive rearrangements occurring in the passage from pre-adolescence to adolescence for the enhancement of social understanding. Furthermore, results from this study evidence how these neuro-psychological advancements in the areas of inhibitory control and cognitive flexibility seem to represent only the maturational scaffolding for the upgrade of social reasoning in this phase of development. The growth of more articulated, flexible, and explicitly mentalistic interpretations of interpersonal exchanges can occur if nurtured by adequate interactive experiences. The affective quality of early as well as current attachment experiences seem to shape the quality of representations of social reality [27]. In particular, the presence of secure or insecure attachment models guides the adolescent’s capacity to explore the epistemic space and intersubjective connections that can be appreciated for the first time of life in this period of maturation [97,98]. Given the relevance of social understanding for adjustment and psychopathology, clinical interventions aiming at improving adolescents’ capacity for mentalization and social reasoning assume a high clinical relevance [12,28]. In particular, a specific focus of such interventions should include affective and communicative transactions within the family context. Targets of family intervention should include supporting the parent to work as a secure base for the adolescent [29], facilitation of the expression and mutual recognition of individual affective needs, and the creation of an awareness within the family of the spaces of sharing as well as of differentiation for each one’s point of view [25,99]. It is important to highlight that insecure attachments are in any case associated with low reflective functioning and the risk of finding themselves in violent relationships both in adolescence and in adulthood: these aspects also influence the mentalization functions and the social cognition skills [100].

Some limitations of this study need to be addressed. For instance, the sample size should be improved, and moreover, none of the participants present a type D or unresolved attachment, which is mostly associated in adolescents with limitations in executive functions and behavioral control [101].

## Figures and Tables

**Table 1 ijerph-20-02836-t001:** Descriptive Statistics for Executive function and Social cognition dimensions.

	11–12 Years Old		13–15 Years Old		Male		Female	
	*N* = 43		*N* = 57		*N* = 50		*N* = 50	
Dimensions	M	SD	M	SD	M	SD	M	SD
Stroop Test	79.80	28.54	61.33	16.41	69.82	22.62	68.73	25.77
CNT	0.31	0.12	0.53	0.23	0.45	0.23	0.41	0.19
Social Cognition (TAT)	2.51	0.34	2.89	0.42	2.64	0.43	2.82	0.43

**Table 2 ijerph-20-02836-t002:** Attachment classification: sex and age differences.

	11–12 Years Old	13–15 Years Old	Male	Female
	*N* = 43	*N* = 57	*N* = 50	*N* = 50
Attachment Classification	f	f	f	f
Secure	22	45	32	35
Preoccupied	5	5	6	4
Dismissing	16	7	12	11

**Table 3 ijerph-20-02836-t003:** Descriptive Statistics and Significant Attachment Classification Group Effects on the Executive function and Social cognition dimensions.

	Secure (*N* = 67)	Preoccupied (*N* = 10)	Dismissing (*N* = 23)
Stroop Test	64.51 ^a^ (17.76)	65.20 ^a^ (19.78)	84.91 ^b^ (34.38)
CNT	0.45 (0.22)	0.41 (0.17)	0.40 (0.21)
Social Cognition (TAT)	2.78 ^a^ (0.42)	2.98 ^a^ (0.47)	2.45 ^b^ (0.33)

Post hoc comparisons (Tukey Test; *p* < 0.05): Different letters indicate mean differences between groups.

**Table 4 ijerph-20-02836-t004:** Zero Order Correlation between Verbal IQ, attachment state of mind dimensions, executive function and social cognition dimensions.

	1	2	3	4	5	6	7	8	9	10	11
1. Verbal IQ	-										
2. Stroop Test	0.04	-									
3. CNT	0.007	−0.41 **	-								
4. Social cognition	−0.13	−0.35 **	0.29 **	-							
5. Lack of Memory	0.03	0.16	0.12	−0.43 **	-						
6. Passivity	0.06	−0.09	−0.07	0.25 *	−0.32 **	-					
7. Coherence of Mind	−0.010	−0.28 **	0.08	0.28 **	−0.54 **	−0.31 **	-				
8. Coherence of transcript	−0.01	−0.27 **	0.08	0.29 **	−0.53 **	−0.32 **	0.99 **	-			
9. Anger	−0.054	−0.14	−0.05	0.16	−0.20 *	0.53 **	−0.32 **	−0.33 **	-		
10. Derogation	0.04	−0.18	0.02	−0.05	0.07	0.13	−0.20 *	−0.20 *	0.34 **	-	
11. Idealization	−0.09	0.37 **	−0.09	−0.46 **	0.67 **	−0.28 **	−0.65 **	−0.64 **	−0.15	0.04	-

Note: * *p* < 0.05, ** *p* < 0.01.

## Data Availability

Data are available on request at the corresponding author mail address.

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
