# Peer review of "Relationship between Executive Functions, Social Cognition, and Attachment State of Mind in Adolescence: An Explorative Study"

_ijerph, 2023, doi:10.3390/ijerph20042836_

Round 1
Reviewer 1 Report
Thank you for asking me to review the manuscript entitled "Relationship Between Executive Functions, Social Cognition, and Attachment State of Mind in Adolescence: An Explorative Study”. I think that the topic is scientifically and clinically relevant, as well as the study is valuable. I have some minor suggestions for the authors that I think may help improve the quality of the contribution before being published.
1- Please, amend the manuscript for several typo errors.
2- Please, the whole manuscript needs a careful proofreading for the English form. Specifically, many sentences are too long and difficult to understand.
3- Page 3, line 115 “….capacity to attribute mental states and interpret behaviors in a mentalistic perspective (40).” Maybe authors could add this paper DOI: 10.1177/0308575914543235 in the text and in the reference list.
4- Page 4, line 157: “Davidson, Amso, Anderson, and Diamond” should be reported as “Davidson et al.” or “Davidson and colleagues.”
5- Page 4, line 16: In the Aims section, it is unclear the meaning of your objective (a): “To provide a quantitative measure of the complexity of social reasoning”. In the results section, I cannot find any explicit answer to the objective (a). Maybe authors should improve clarity of their aims explaining their research questions.
6- Page 5, line 230: “in the manuals developed by (77)”, Please, amend the sentence adding the name of the first author of the study numbered 77.
7- Page 5, line 225-226: “Social Cognition. The Social Cognition and Object Relations Scale (SCORS; 77) was used to evaluate participants’ quality of object relations from TAT responses”. Authors have indicated that the SCORS assessment is applied on the responses to the Thematic Apperception Test. Albeit TAT scores are not considered in the study, this test should be briefly described to understand the nature of participants responses. Please add a short explanation of this instrument.
8- Page 6, lines 275-276: “Verbal ability. We used the Vocabulary subtest from the Italian version of WISC III (81) to assess word knowledge, language development and long-term memory.” Please provide more details about the Vocabulary test.
9- Page 6, lines 298: there is a typo, please change “Kolmogorv-Smirnov” in “Kolmogorov-Smirnov”
10- Page 7, line 318: “The distribution of the four attachment classification groups is presented in Table 1.” I suppose that the authors refer to Table 2 (not 1). Moreover, only three attachment dimensions are reported in Table 2 (frequencies of Unresolved attachment are missing, maybe because they are zero). Please, amend the sentence.
11- Page 7, line 324: “78.9% of 13-15 years old vs. 51.2 % of males”. Maybe “males” should be “11-12 years old”.
12- Discussion (Page 9, lines 377-379): “This evidence may be considered as indirect evidence of the validity of the assessment of social understanding carried out in his study”. Authors should remove this statement because the gender differences cannot be considered an index of validity of any instrument. I suggest saying that your results are in line with previous evidence about gender differences in social cognition skills.
13- The Discussion made causal interpretations of the correlational relationships emerged (e.g., “dismissing attachment predicts significantly lower scores in the social cognition measure”). This is misleading since you have only correlational results, and causal inferences cannot be done.
14- Page 9-10, lines 416-438, the Discussion in which the issues connected to the dismissing pattern are commented can me enriched by inserting this paper DOI: 10.1007/s10826-021-02181-1
Author Response
1- we thank you for your comment and we corrected the typos we found
2- we agree with your comment and we shortened some of the longest sentences
3- we thank you for your advice and we added the suggested paper
4- we changed in "Davidson and colleagues."
5- We agree that we had to better formulate the original objectives and reformulated them into two specific questions. We also agree that "the creation of a new reliable valid measure of adolescent social reasoning" was not a research question in itself; nonetheless we decided to maintain the part of the discussion that analyze the use of this measure, give the shortage of empirical assessment of social reasoning in adolescence
6- we corrected this typo
7- line 236-37 we provided a short description of the way SCORES were applied to TAT tables.
8- we provided the requested informations: This task measures lexical knowledge and, more specifically, the ability to retrieve a word’s meaning and to provide an accurate definition. Participants were asked to orally define 30 stimulus words presented by the examiner; the words increased in difficulty and abstraction as the test progressed. Each child received a score of 0, one or two points for each item based on the quality of the definition given. The score can range from 0 to 60.
9- we corrected the typo
10- we amended the sentence
11- we corrected the typo
12- we thank you for your suggestion, and we followed it
13- we agree with you concern and we modified according to it (e.g. line 423)
14- we inserted the suggested reference

Reviewer 2 Report
the article is very interesting and clear to read. I would only suggest small revisions. I advise the authors to further highlight the limitations of their study: the sample is small and, moreover, none of them presents a type D or unresolved attachment that is mostly associated in adolescents with limitations in executive functions and behavioral control. (Gallarin, M.; Torres-Gomez, B.; Alonso-Arbiol, I. Aggression in Adopted and Non-Adopted Adolescents: The Role of Parenting, Attachment Security, and Gender. Int. J. Environ. Ris. Salute publishes 2021, 18, 2034. https://doi.org/10.3390/ijerph18042034) I also suggest that the authors highlight how in any case insecure attachments are associated with low Reflective functioning and the risk of finding themselves in violent relationships both in adolescence and in adulthood (Condino et al. 2022. Attachment, Trauma, and Mentalization in Intimate Partner Violence: A Preliminary Investigation. Journal of Interpersonal Violence; Bonache, H., Gonzalez-Mendez, R. & Krahé, B. Romantic Attachment, Conflict Resolution Styles, and Teen Dating Violence Victimization. J Youth Adolescence 46, 1905–1917 (2017).These aspects influence the mentalization functions and the social cognition skills
Author Response
Dear colleague, thank you very much for your helpful insights, we appreciated and followed you suggestions as you can see from line 481 to 489
